# THE KNN SCORE FOR EVALUATING PROBABILISTIC MULTIVARIATE TIME SERIES FORECASTING

## ABSTRACT

Time series forecasting is a critical task in various domains. With the aim of comprehending interconnections and dependencies among variables, as well as gaining insights into a range of potential future outcomes, probabilistic multivariate time series forecasting has emerged as a prominent approach. The evaluation of models employed in this task is crucial yet challenging. Comparing a set of predictions against a single observed future presents difficulties, and accurately measuring whether a model correctly predicts dependencies between different time steps and individual series further compounds the complexity. We observe that metrics which are currently employed fall short in providing a comprehensive assessment of model performance. To address this limitation, we propose a novel metric based on density estimation as an alternative. We showcase the advantages of our metric both qualitatively and quantitatively, underscoring its effectiveness in assessing forecast quality.

## 1 INTRODUCTION

Time series forecasting is of paramount importance in various domains. It involves predicting future values based on historical data, offering insights into trends, patterns, and potential outcomes. Univariate models have historically been the predominant choice. These models aim to predict the future values of a single variable based on its historical data. While they have been widely used, univariate forecasting falls short in capturing the intricate relationships and dependencies that exist among multiple variables.

Furthermore, providing only a single point estimate as a prediction, without considering uncertainty, has been the norm in traditional time series forecasting. These non-probabilistic models provide a singular future prediction without quantifying the range of potential outcomes and their associated probabilities. However, decision-makers often require a more comprehensive understanding of the potential scenarios to make well-informed choices. Probabilistic time series forecasting addresses this limitation by providing a range of potential future outcomes.

The task of multivariate probabilistic time series forecasting surpasses the limitations of univariate forecasting and non-probabilistic approaches. By considering the relationships among multiple variables and incorporating probabilistic modeling, one can achieve a more comprehensive understanding of system dynamics while providing uncertainty estimates. This makes it possible to improve forecasting accuracy, to enhance risk assessment capabilities, and to facilitate effective decision-making in a wide range of applications Gneiting & Raftery (2007).

It is essential but difficult to evaluate the effectiveness of multivariate forecasts. Most current works utilize CRPS and CRPS-Sum (e.g. Drouin et al. (2022); Kan et al. (2022); Rasul et al. (2021b)) or MSIS (e.g. Gasthaus et al. (2019); Gouttes et al. (2021); Park et al. (2022)) to evaluate proposed multivariate probabilistic models against previous approaches. As these are univariate evaluation techniques, they fail to measure dependencies between different time steps or different series. Further metrics like the Energy Score (Gneiting & Raftery, 2007) and the Variogram Score (Scheuerer & Hamill, 2015) have been proposed as alternatives, but are not used as often. While both of are multivariate scoring rules, they have drawbacks in different aspects. The Energy Score is insensitive to correlation differences (Pinson & Tastu, 2013; Alexander et al., 2022; Scheuerer & Hamill, 2015) and the Variogram Score is not rotation invariant.

In this paper, we show a comprehensive analysis of prior metrics, highlighting their shortcomings and limitations. Building upon this critical examination, we introduce an alternative metric, the $k$-Nearest Neighbor Score, grounded in density estimation. We assess our proposed metric thoroughly and demonstrate its capabilities using synthetic data and deep learning models trained on real datasets. Our experiments show that our metric is a viable alternative for evaluating models for multivariate time series predictions.

## 2 METRICS FOR TIME SERIES

Numerous metrics have been proposed to assess the accuracy of forecasts for multivariate time series data. In this section, we will give a comprehensive review and discussion of these evaluation metrics.

A multivariate time series can be denoted as a matrix $\boldsymbol{X} \in \mathbb{R}^{D \times T}$, where an element $X_{d,t}$ describes the value of dimension $d$ at time $t$. For multivariate time series prediction, we have access to historic values $\boldsymbol{Y}^-$ and aim to forecast the future values $\boldsymbol{Y}$. A probabilistic forecasting method learns a probability distribution $p(\boldsymbol{X})$ over future predictions. Consistent with previous work (Gneiting & Raftery, 2007; Scheuerer & Hamill, 2015), we represent this distribution nonparametrically, as a set of $N$ future predictions $\boldsymbol{X}^i \in \mathbb{X}$. A metric for time series needs to evaluate such a prediction set $\mathbb{X}$ given the true future $\boldsymbol{Y}$.

### 2.1 PROPRIETY

An important property of a metric for probabilistic forecasts is propriety. Following Gneiting & Raftery (2007), a metric $S(P, y)$ with prediction $P$ and observed future $y$ is proper if

$$S(Q, Q) \geq S(P, Q) \quad \forall P, Q, \tag{1}$$

where

$$S(P, Q) = \int S(P, y) dQ(y). \tag{2}$$

A metric is strictly proper if equality in Equation 1 only holds if $P = Q$.

Intuitively, a metric is considered proper when a model that accurately predicts the true data distribution receives the highest score possible. When samples are predicted instead of an entire distribution, optimizing the metric should involve generating samples that closely match the data distribution. While propriety is crucial, it does not inherently provide insights into their practical utility for evaluating less-than-perfect models. To illustrate this point, consider a hypothetical oracle metric that assigns all models identical scores, except for those that perfectly mimic the true data distribution. Such a metric would indeed be strictly proper but ultimately serve no meaningful purpose.

### 2.2 METRICS FOR POINT FORECASTS

Many metrics are available for univariate non-probabilistic settings, where a forecasting method only outputs a single prediction $\boldsymbol{x}$. Often, each individual $x_t$ is compared with the observed future $y_t$ by computing the absolute error $|x_t - y_t|$ or the squared error $(x_t - y_t)^2$. The results are then optionally weighted, scaled and averaged over the individual series and time steps. Common examples are the symmetric mean average percentage error (Armstrong, 1985)

$$\text{sMAPE}(\boldsymbol{x}, \boldsymbol{y}) = \frac{200}{T} \sum_{t=1}^{T} \frac{|x_t - y_t|}{|y_t| + |x_t|} \tag{3}$$

and the mean absolute scaled error (Hyndman & Koehler, 2006)

$$\text{MASE}(\boldsymbol{x}, \boldsymbol{y}) = \frac{1}{T} \frac{\sum_{t=1}^{T} |x_t - y_t|}{\frac{1}{T-1} \sum_{t=2}^{T} |y_t - y_{t-1}|}. \tag{4}$$

Alternatives are the mean absolute error (MAE), the mean squared error (MSE), the root mean squared error (RMSE), the normalized root mean squared error (NRMSE), the mean absolute percentage error (MAPE) and the weighted absolute percent error (WAPE) (Hyndman & Athanasopoulos, 2018).

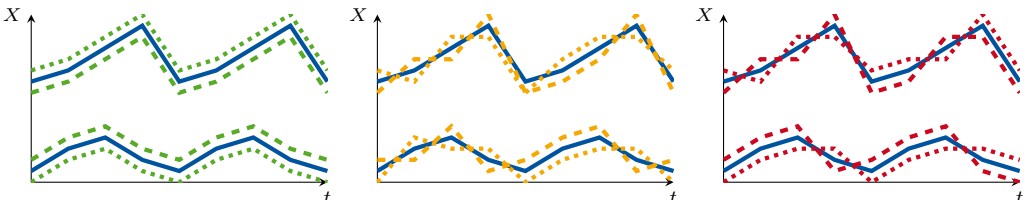

Figure 1: Time series with predictions. Visualized is the true future of a bivariate time series (blue) and two predictions (dashed/dotted) of three models (green/orange/red). Only the green one mimics dependencies between time steps correctly. CRPS and MSIS rate all forecasts the same as the distribution around each individual point is the same for all models. CRPS-Sum would be optimal for the green and orange forecast, because the sum of the predictions equals the observed series.

These metrics have been applied for the evaluation of probabilistic models (e.g. Nguyen & Quanz (2021); Rasul et al. (2021a;b)) despite their inability to evaluate probabilistic predictions. In fact, optimizing such metrics directly leads to a variant of either mean regression or median regression (for squared or absolute error, respectively). Consequently, a model which is optimal under such a metric produces the mean or median of the data distribution instead of providing a genuine probabilistic forecast. As a result, these metrics should be avoided when assessing probabilistic predictions.

## 2.3 CONTINUOUS RANKED PROBABILITY SCORE

The Continuous Ranked Probability Score (Brown, 1974; Gneiting & Raftery, 2007) is a proper metric and compares a one-dimensional forecast given by a cumulative distribution function $F$ with the true future scalar $y$. It is defined as

$$\text{CRPS}(F, y) = -\int_{-\infty}^{\infty} (F(x) - \mathbb{1}\{x \geq y\})^2 dx. \tag{5}$$

The empirical cumulative distribution function $\hat{F}(x) = \frac{1}{|\mathbb{X}|} \sum_{x' \in \mathbb{X}} \mathbb{1}\{x' \leq x\}$ can be applied if the prediction is given by an ensemble of points $\mathbb{X}$. Then, CRPS can be evaluated as

$$\text{CRPS}(\mathbb{X}, y) = \frac{1}{2} \mathbb{E}_{\substack{x,x' \in \mathbb{X} \\ x \neq x'}} |x - x'| - \mathbb{E}_{x \in \mathbb{X}} |x - y|. \tag{6}$$

CRPS is applied on multivariate time series by comparing each observed value $Y_{d,t}$ to the predictions $X_{d,t}^i$ for $i \in [1, K]$. The results are aggregated by averaging across all dimensions and time steps. Since each series and each time step is evaluated independently, CRPS cannot capture dependencies between them. Nevertheless it has been applied for multivariate time series evaluation (e.g. Drouin et al. (2022); Gouttes et al. (2021); Kan et al. (2022); Park et al. (2022); Rasul et al. (2021b); Salinas et al. (2019); Tashiro et al. (2021)).

As a case in point, Figure 1 visualizes a bivariate time series and three models, each with two predictions. While the predictions from the first model only differ by a constant shift from the observation, the other two models completely fail to capture the temporal dependencies. However, the predictive distribution for each time step and each individual series is the same in all cases, thus all these models receive an identical CRPS.

CRPS-Sum (Salinas et al., 2019) is an adaption of CRPS and has emerged as a main metric for multivariate settings (e.g. de Bézenac et al. (2020); Drouin et al. (2022); Kan et al. (2022); Nguyen & Quanz (2021); Rasul et al. (2021a;b); Salinas et al. (2019); Tang & Matteson (2021); Tashiro et al. (2021)). The individual time series are summed together to form a single, univariate time series $x$ with $x_t = \sum_{d=1}^{D} X_{d,t}$. Then, CRPS is applied for each time step separately.

This unveils certain dependencies between the different time series for the metric, but major issues emerge. To begin, the evaluation process still treats each time step in isolation, such that correlation between different time steps remains hidden. Second, opposing characteristics of individual time series might cancel out if they are summed. If $Y_{0,t} = -Y_{1,t} \forall t$, a model predicting $X_{0,t} = X_{1,t} = 0$ would achieve an optimal score. Also, adding element-wise Gaussian noise to the forecast would not harm the rating. Lastly, the metric loses all information about the individual time series, such that predicting the sum of the series is sufficient for an optimal rating.

The sum of the orange prediction and the green prediction in Figure 1 is the same and it even equals the sum of the true future, so both models would receive the optimal score. The red model has a different CRPS-Sum, since the series' sum is different.

## 2.4 MEAN SCALED INTERVAL SCORE

The Interval Score (Gneiting & Raftery, 2007) makes use of quantiles of the predicted distribution. These can be estimated from a set of predictions $\mathbb{X}$ if the model cannot output quantiles by itself. It is defined as

$$\text{IS}_\alpha(\mathbb{X}, y) = (u - \ell) + \frac{2}{\alpha}(\ell - y)\mathbb{1}\{y < \ell\} + \frac{2}{\alpha}(y - u)\mathbb{1}\{y > u\}, \tag{7}$$

where $\ell$ and $u$ define the $\frac{\alpha}{2}$ and $1 - \frac{\alpha}{2}$ quantiles, respectively.

The Mean Scaled Interval Score (Makridakis et al., 2020) averages the Interval Score and normalizes it with the mean absolute seasonal difference of the series history which corresponds to the mean absolute error of a naïve forecaster. Let $m$ be the length of a season and $\tau$ be the number of historic values $y_t^-$, then

$$\text{MSIS}_\alpha(\mathbb{X}, y) = \frac{\frac{1}{T}\sum_{t=1}^{T} \text{IS}_\alpha(\mathbb{X}_t, y_t)}{\frac{1}{T-m}\sum_{t=m+1}^{\tau} |y_t^- \, y_{t-m}^-|}. \tag{8}$$

MSIS has been applied by various authors, especially for models which output quantiles instead of predicting ensembles (e.g. Gasthaus et al. (2019); Gouttes et al. (2021); Kan et al. (2022); Park et al. (2022)). However, it suffers from the same issues as CRPS: the individual evaluation of each time step and each series does not allow to measure dependencies between them. All examples in Figure 1 would again be scored the same.

## 2.5 ENERGY SCORE

The Energy Score (Gneiting & Raftery, 2007) is a straightforward generalization of the CRPS formulation in Equation 6 to multiple dimensions, where the absolute difference is replaced by the Euclidean distance. With $p \in (0, 2)$, the Energy Score is defined as

$$\text{ES}_p(\mathbb{X}, \boldsymbol{y}) = \frac{1}{2}\mathbb{E}_{\substack{\boldsymbol{x}, \boldsymbol{x}' \in \mathbb{X} \\ \boldsymbol{x} \neq \boldsymbol{x}'}} \|\boldsymbol{x} - \boldsymbol{x}'\|^p - \mathbb{E}_{\boldsymbol{x} \in \mathbb{X}} \|\boldsymbol{x} - \boldsymbol{y}\|^p. \tag{9}$$

Intuitively, the second term of this metric computes the average Euclidean distance between the real observation $\boldsymbol{y}$ and the predictions $\boldsymbol{x} \in \mathbb{X}$. It is minimized by the geometric median (Cohen et al., 2016), thus an optimal predictor would only forecast this point instead of a distribution. The first term counters this behavior by rewarding a better score if the model gives diverse predictions.

When applying the Energy Score on multivariate time series, both the observed future $\boldsymbol{Y}$ and the predictions $\boldsymbol{X}$ are flattened to vectors $\boldsymbol{y}$ and $\boldsymbol{x}$, respectively. This way, the Energy Score allows measuring the dependencies between both different time series and different time steps. The asymptotic runtime is $\mathcal{O}(N^2 DT)$ because the Energy Score computes the distances between each pair of predictions. As the evaluation quality increases for bigger $N$, this is a relevant disadvantage. So far, the Energy Score has been applied only sparsely (Drouin et al., 2022; Muniain & Ziel, 2020; Kan et al., 2022; Dumas et al., 2022).

### 2.5.1 RELATION TO CRPS

We now present an interesting property of the Energy Score. See Appendix A for the proof.

**Theorem 1.** *Let $\boldsymbol{r}$ be a $d$-dimensional random vector with unit length. Then there exists a constant $c_d$, such that*

$$\mathbb{E}_{\boldsymbol{r}}\text{CRPS}(\{\boldsymbol{r} \cdot \boldsymbol{x} : \boldsymbol{x} \in \mathbb{X}\}, \boldsymbol{r} \cdot \boldsymbol{y}) = c_d \cdot \text{ES}_1(\mathbb{X}, \boldsymbol{y}). \tag{10}$$

According to Theorem 1, applying CRPS on random projections of the data and then averaging corresponds to the Energy Score. Since all marginals are evaluated independently, certain properties of the distribution are lost. Figure 2 shows a toy example with a circular prediction. The Energy Score is optimal, if the observation lies in the center of the forecast, even though none of the predictions are remotely close. A forecast with some predictions close to the true data point scores worse.

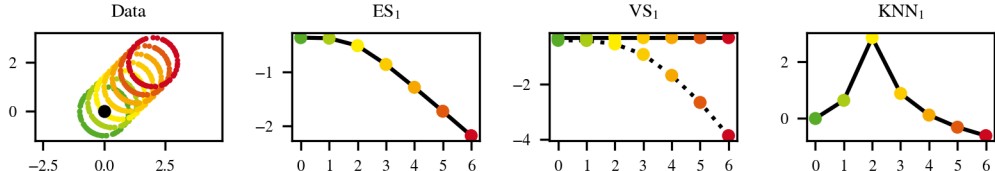

Figure 2: Multivariate scores on circular forecasts. Energy Score, Variogram Score and KNN Score are evaluated for different shifts. Only the yellow forecast has some predictions close to the observation (black). If predictions and observation are rotated by $45°$, the Variogram Score produces a different rating (dotted line), Energy Score and KNN Score remain the same if rotated.

## 2.6 VARIOGRAM SCORE

The Variogram Score (Scheuerer & Hamill, 2015) is a proper metric which computes the pairwise differences between the variables and compares them to the observed differences:

$$\text{VS}_p(\mathbb{X}, \boldsymbol{y}) = \sum_{i=1}^{d} \sum_{j=1}^{d} \left( |y_i - y_j|^p - \mathbb{E}_{\boldsymbol{x} \in \mathbb{X}} \left( |x_i - x_j|^p \right) \right)^2. \tag{11}$$

Scheuerer and Hamill already note, that the Variogram Score is not able to notice if all components are shifted by the same value and that it does not detect if distributions differ in moments which are higher than $p$.

Another downside is the absence of rotational invariance. If both prediction and observation vectors undergo the same rotation, the Variogram Score changes. This is not desirable, because we cannot assume that the directions of the unit vectors hold any special significance. Consider for example a time series that records 2-dimensional positions of a vehicle. It is important that errors along a coordinate axis are treated equally to errors occurring along the diagonal direction, without any bias in valuation. All forecasts in Figure 2 are scored the same by the Variogram Score, but a rotation of $45°$ changes the outcome of the score drastically even though the severity of the errors stays the same.

If the Variogram Score is applied to flattened predictions in order to measure both temporal and inter-series dependencies, the asymptotic runtime is $\mathcal{O}(ND^2T^2)$, which is very costly in many scenarios.

## 3 $k$-NEAREST NEIGHBOR SCORE

We will now present our novel scoring rule for probabilistic multivariate time series forecasting. If a forecasting method allows to evaluate the density $p$ of the forecast at an arbitrary point, the Logarithmic Score $\text{LogS}(p, \boldsymbol{y}) = \log p(\boldsymbol{y})$ (Gneiting & Raftery, 2007) is a valid choice for evaluation. It corresponds to maximum likelihood estimation, which is often used to train probabilistic models like Normalizing Flows (Dinh et al., 2014). In many cases however, probabilistic forecasting methods are not able to evaluate density, they are only able to produce samples from the target distributions.

For this reason, a metric ideally relies on samples only, and thus we propose to combine the Logarithmic Score with $k$-nearest neighbors density estimation (Loftsgaarden & Quesenberry, 1965). After omitting constant terms, we define the $k$-Nearest Neighbor Score as

$$\text{KNNS}_k(\mathbb{X}, \boldsymbol{y}) = -\log \left\| \text{NN}_{\mathbb{X}}^k(\boldsymbol{y}) - \boldsymbol{y} \right\|, \tag{12}$$

where $\text{NN}_{\mathbb{X}}^k(\boldsymbol{y})$ is the $k$-th nearest neighbor of $\boldsymbol{y}$ from the ensemble of predictions $\mathbb{X}$.

The KNN Score behaves as expected for the example in Figure 2. The best score is assigned to the forecast which has an overlap with the true observation. If none of the predictions is close to the observation, the score drops.

For a sufficient number of samples, a density estimate would match the true density sufficiently well, which would make the metric proper. However, a realistic multivariate time series with a few hundred dimensions and a prediction horizon of 20 to 30 time steps would already correspond to a

flattened vector with more than a thousand dimensions. Due to the curse of dimensionality, sampling sufficiently many prediction vectors is practically impossible.

We therefore take inspiration from the Energy Score's correspondence to the projected CRPS (see Section 2.5.1). More precisely, we perform random projections of the true and generated samples to a $d$-dimensional subspace and compute the average distance over multiple projections. Let the row vectors of $\boldsymbol{P} \in \mathbb{R}^{d \times DT}$ be a random orthonormal basis of a $d$-dimensional subspace, then

$$\text{KNNS}_k^d(\mathbb{X}, \boldsymbol{y}) = -\log \mathbb{E}_{\boldsymbol{P}} \left\| \text{NN}_{\{\boldsymbol{Px}|\boldsymbol{x} \in \mathbb{X}\}}^k(\boldsymbol{Py}) - \boldsymbol{Py} \right\|. \tag{13}$$

We create $\boldsymbol{P}$ by sampling the values from a Gaussian and normalizing the rows to ensure unit length. For efficiency, we do not enforce the orthogonality of the rows, but since $d \ll DT$, the rows will almost always be almost orthogonal.

Instead of random projections, one could use dimensionality reduction techniques like principal component analysis (Pearson, 1901), which aim to find a "good" projection to a lower-dimensional space. However, if the reduction is based on the prediction of a model, the model could game the metric by omitting dimensions if it is unsure about them. One could also attempt to find a dimensionality reduction transformation without incorporating forecasts, for example using the test set, but in this case models could already be trained directly on the projected data.

Kernel density estimation (Rosenblatt, 1956; Parzen, 1962) would be an alternative to $k$-nearest neighbors, but it has more parameters to choose from which makes it more difficult to select suitable ones. Apart from the specific kernel, it would be required to also select a bandwidth size and shape. Choosing these automatically based on either prediction or data would lead to similar issues as described in the previous paragraph.

If $\boldsymbol{x}_i, \boldsymbol{y} \in \mathbb{R}^{DT}$, $i \in [1, N]$ and we use $N_{\text{RP}}$ random projections to $d$-dimensional space, the projections can be computed in $\mathcal{O}(N_{\text{RP}}NDTd)$ and all Euclidean distances between $X_i$ and $Y$ in $\mathcal{O}(N_{\text{RP}}Nd)$. If we search for the $k$-th closest neighbor by sorting, this requires an additional $\mathcal{O}(N \log N)$. By using a max-heap of the smallest $k$ elements, it is sufficient to traverse the distances only once while keeping the heap intact. This requires $\mathcal{O}(N \log k)$ time, so $\mathcal{O}(N_{\text{RP}}NDTd + N \log k)$ in total for the KNNS Score. Both $d$ and $k$ are small and the process can be parallelized quite well, so the total runtime is manageable.

## 4 EVALUATING METRICS

Assessing the quality of score functions always follows a similar idea. Define a true data generating model $p_0$ and an adapted model $p_1$. Then, sample an observation $\boldsymbol{y} \sim p_0$ and two forecast ensembles $\mathbb{X}_0$ and $\mathbb{X}_1$ from $p_0$ and $p_1$ respectively. On average, we expect the forecast from the true model to score higher than the forecast of the wrong model. For proper metrics this is true if $|\mathbb{X}| \to \infty$, but due to the finite sampling and the finite size of datasets this is often not fulfilled in practice.

Pinson & Tastu (2013) examine changes in mean, variance and correlation for two-dimensional Gaussians and compute the relative score difference between a given model and the true data generating model for the Energy Score. Even in this simple setting, they observe that the Energy Score is much less sensitive to changes in the correlation structure.

To underpin the validity of their Variogram Score, Scheuerer & Hamill (2015) perform similar experiments with Gaussians of up to 15 dimensions. Apart from the Energy Score, they also compare to the Dawid-Sebastiani Score (Dawid & Sebastiani, 1999), which assumes Gaussian distributions. They define a default Gaussian distribution and a set of adapted Gaussians with different means, different variances or different correlation. This results in a distribution of scores for each adapted model, visualized by a boxplot, and they argue, that a metric has a low discrimination ability if the boxplots for the adapted models overlap strongly with the boxplots of the real model.

This way of measuring discrimination ability has an issue. Assume we have two forecasts $\boldsymbol{y}, \boldsymbol{y}' \sim p_0$ and ensembles $\mathbb{X}_0$ from the true distribution and $\mathbb{X}_1$ from a different model. We would expect that $S(\boldsymbol{y}, \mathbb{X}_0) \geq S(\boldsymbol{y}, \mathbb{X}_1)$, but whether $S(\boldsymbol{y}, \mathbb{X}_0) \geq S(\boldsymbol{y}', \mathbb{X}_1)$ holds as well does not matter. In real scenarios only a single observation is available, thus only the score difference to the same observation is relevant.

Alexander et al. (2022) compare the Energy Score with the Variogram Score using models trained on real data. They propose a "generalized discrimination heuristic" which also suffers from the above problem due to averaging of scores using different observations. It further computes a relative score, i.e. the average score of $p_1$ divided by the average score of $p_0$. This is problematic because a metric would be penalized if $S(\boldsymbol{y}, \mathbb{X}_0)$ is high in general, even though this does not lower the discrimination ability. However, they also visualize the distribution of differences $S(\boldsymbol{y}, \mathbb{X}_1) - S(\boldsymbol{y}, \mathbb{X}_0)$, which allows to compare scores of different models on the same observation $\boldsymbol{y}$.

These papers share the consensus that the Energy Score exhibits limitations in capturing correlation structure effectively. In contrast, the Variogram Score demonstrates better performance in addressing these situations. In the following section, we will show how our KNN Score behaves in comparison.

## 5 EXPERIMENTS

We roughly follow Alexander et al. (2022) to compute the discrimination ability of a metric $S$ and use the difference $S(\boldsymbol{y}, \mathbb{X}_0) - S(\boldsymbol{y}, \mathbb{X}_1)$ to measure how well the scoring rule $S$ discriminates a model $p_1$ from the data generating process $p_0$. We simulate a test dataset by averaging this difference for multiple forecasts and prediction sets. Finally, we repeat this process multiple times to collect a distribution of results. A metric is considered more effective at distinguishing between $p_0$ and $p_1$ when these results are greater than 0 more frequently.

Alexander et al. (2022) visualize these distributions as density plots, Scheuerer & Hamill (2015) use boxplots. These are hard to compare for a larger amount of experiments, so we convert them into two discrimination scores. Let $\mathbb{D}$ be the distribution of averaged differences and $\mathbb{D}^- = \{d \in \mathbb{D} \mid d < 0\}$. Then,

$$\mathrm{DS}_c = \frac{|\mathbb{D}^-|}{|\mathbb{D}|} \qquad \text{and} \qquad \mathrm{DS}_a = \frac{-\sum_{d \in \mathbb{D}^-} d}{\sum_{d \in \mathbb{D}} |d|}. \tag{14}$$

Therefore, $\mathrm{DS}_c$ measures the ratio of negative results, whereas $\mathrm{DS}_a$ aggregates the results while taking their actual values into consideration.

### 5.1 SYNTHETIC DATA

We sample prediction ensembles of size 100, average the differences over 1000 observations and repeat these experiments 50 times to get the distribution of results. We experiment with $\mathrm{ES}_p$ for $p \in \{0.5, 1, 1.5\}$ and $\mathrm{VS}_p$ for $p \in \{0.5, 1, 1.5, 2\}$. For our $\mathrm{KNNS}_k^d$ Score, we test $k, d \in \{1, 2, 3, 4, 5\}$ and use $N_{\mathrm{RP}} = 1000$ repetitions.

We use similar settings as Scheuerer & Hamill (2015) for unimodal experiments, but we increase the dimension to 100, as dimensions of 5 or 15 are far from realistic scenarios. The default model $p_0$ is a 100-dimensional Gaussian $\mathcal{N}(\boldsymbol{0}, \boldsymbol{\Sigma})$ with correlation function $\exp\left(-\frac{|i-j|}{3}\right)$. The following adapted models are considered: (a) mean bias: the values of $\boldsymbol{\mu}$ are linearly interpolated between $\mu_1 = -0.25$ and $\mu_{100} = 0.25$; (b) larger variance: the variance is increased to 1.5; (c) smaller variance: the variance is decreased to $\frac{2}{3}$; (d) less correlation: the correlation function is $\exp\left(-\frac{|i-j|}{2}\right)$; (e) more correlation: the correlation function is $\exp\left(-\frac{|i-j|}{4.5}\right)$; (f) correlation model (i): the correlation function is $\left(1 + \frac{|i-j|}{3}\right)^{-1}$; (g) correlation model (ii): the correlation function is $\exp\left(-\frac{|i-j|}{4}\right)\left[\frac{3}{4} + \frac{1}{4}\cos\left(\frac{|i-j|\pi}{2}\right)\right]$.

Table 1 shows $\mathrm{DS}_a$ for a subset of experiments, full tables are in Appendix B. The Energy Score has issues with distinguishing correlation in general, which has been noted in previous works (Pinson & Tastu, 2013; Scheuerer & Hamill, 2015; Alexander et al., 2022). Since the Variogram Score was proposed to tackle this issue, it does perform well in these cases. It has the most issues with the mean bias, even though this bias is different for all components.

The KNN Score performs best for $d = 2, k \in \{2, 3\}$, where it is better than the Energy Score and on par with the Variogram Score. For $k = 1$, it cannot distinguish $p_0$ from the model with larger variance and the one with less correlation, for $k \geq 3$ the opposite behavior emerges and it scores the stronger correlated and less variable model better than $p_0$.

Table 1: $DS_a$ for unimodal experiments.

| | mean bias | larger var | smaller var | less corr | more corr | corr (i) | corr (ii) |
|---|---|---|---|---|---|---|---|
| $ES_{0.5}$ | 0.00 | 0.00 | 0.00 | 0.34 | 0.18 | 0.02 | 0.29 |
| $ES_1$ | 0.00 | 0.00 | 0.00 | 0.43 | 0.29 | 0.05 | 0.37 |
| $ES_{1.5}$ | 0.00 | 0.00 | 0.00 | 0.53 | 0.42 | 0.22 | 0.47 |
| $VS_{0.5}$ | 0.06 | 0.00 | 0.00 | 0.00 | 0.00 | 0.00 | 0.00 |
| $VS_1$ | 0.05 | 0.00 | 0.00 | 0.00 | 0.00 | 0.00 | 0.00 |
| $VS_{1.5}$ | 0.04 | 0.00 | 0.00 | 0.00 | 0.01 | 0.00 | 0.01 |
| $VS_2$ | 0.03 | 0.00 | 0.00 | 0.01 | 0.04 | 0.00 | 0.04 |
| $KNNS_1^2$ | 0.00 | 1.00 | 0.00 | 0.11 | 0.00 | 0.00 | 0.04 |
| $KNNS_2^2$ | 0.00 | 0.00 | 0.00 | 0.04 | 0.01 | 0.00 | 0.04 |
| $KNNS_3^2$ | 0.00 | 0.00 | 0.00 | 0.01 | 0.06 | 0.00 | 0.03 |
| $KNNS_4^2$ | 0.00 | 0.00 | 0.00 | 0.00 | 0.23 | 0.00 | 0.02 |
| $KNNS_5^2$ | 0.00 | 0.00 | 0.00 | 0.00 | 0.46 | 0.00 | 0.02 |

Table 2: $DS_a$ for multimodal experiments.

| | 3 components | | | | 10 components | | | |
|---|---|---|---|---|---|---|---|---|
| | mean distance | | weights | | mean distance | | weights | |
| | larger | smaller | similar | different | larger | smaller | similar | different |
| $ES_{0.5}$ | 0.16 | 0.27 | 0.44 | 0.37 | 0.27 | 0.48 | 0.46 | 0.53 |
| $ES_1$ | 0.17 | 0.28 | 0.44 | 0.37 | 0.27 | 0.48 | 0.45 | 0.53 |
| $ES_{1.5}$ | 0.17 | 0.28 | 0.44 | 0.38 | 0.26 | 0.48 | 0.45 | 0.52 |
| $VS_{0.5}$ | 0.00 | 0.52 | 0.51 | 0.33 | 0.04 | 0.52 | 0.57 | 0.45 |
| $VS_1$ | 0.00 | 0.53 | 0.55 | 0.33 | 0.02 | 0.54 | 0.57 | 0.43 |
| $VS_{1.5}$ | 0.00 | 0.54 | 0.57 | 0.33 | 0.01 | 0.56 | 0.57 | 0.42 |
| $VS_2$ | 0.00 | 0.54 | 0.58 | 0.33 | 0.01 | 0.58 | 0.56 | 0.40 |
| $KNNS_1^2$ | 0.45 | 0.47 | 0.57 | 0.46 | 0.86 | 0.35 | 0.40 | 0.51 |
| $KNNS_2^2$ | 0.31 | 0.39 | 0.53 | 0.39 | 0.70 | 0.38 | 0.40 | 0.44 |
| $KNNS_3^2$ | 0.24 | 0.34 | 0.49 | 0.38 | 0.53 | 0.41 | 0.41 | 0.45 |
| $KNNS_4^2$ | 0.20 | 0.35 | 0.48 | 0.38 | 0.38 | 0.42 | 0.41 | 0.46 |
| $KNNS_5^2$ | 0.17 | 0.35 | 0.46 | 0.38 | 0.27 | 0.47 | 0.43 | 0.46 |

So far we have only considered unimodal Gaussians. We will now use Gaussian mixture models using the same correlation structure as before for all components. The model $p_0$ has $N_C$ components and we set the component weights proportional to $\pi_i = i$. We define the mean $\boldsymbol{\mu}_i$ of each component $i$ to be $(\boldsymbol{\mu}_i)_j = \frac{i}{N_C}\mathbb{1}\{i = j\}$, i.e. value $i$ of component $i$ is $\frac{i}{N_C}$, all others are 0. We look at these four adaptions: (a) larger mean distance: $(\boldsymbol{\mu}_i)_j = \frac{2i}{N_C}\mathbb{1}\{i = j\}$; (b) smaller mean distance: $(\boldsymbol{\mu}_i)_j = \frac{0.5i}{N_C}\mathbb{1}\{i = j\}$; (c) more similar component weights: proportional to $\pi_i = \sqrt[3]{i}$; (d) more different component weights: proportional to $\pi_i = i^3$.

We perform these experiments for $N_C \in \{3, 10\}$, results are shown in Table 2. KNNS achieves the highest discrimination ability for $d = 2, k \in \{3, 4\}$. Increasing $k$ or $d$ leads to a better separability between $p_0$ and the model with components further away, but has the opposite effect if components are pushed together. If the component weights are changed, the selection of $k$ and $d$ does not have much effect. When compared to $KNNS_2^4$, VS performs better for the first model, worse for the second and third and slightly better for the last. ES is worse in all cases except for comparing $p_0$ with 10 components to the model with larger mean distances.

## 5.2 REAL DATA

Instead of synthesizing artificial data from Gaussians or from mixture models, we can also use deep learning models for time series forecasting, train them on real data and use one of them as the data generating process. We follow the training and evaluation procedure described by Drouin et al. (2022), as they provide code and detailed hyperparameters for six models on five different datasets.

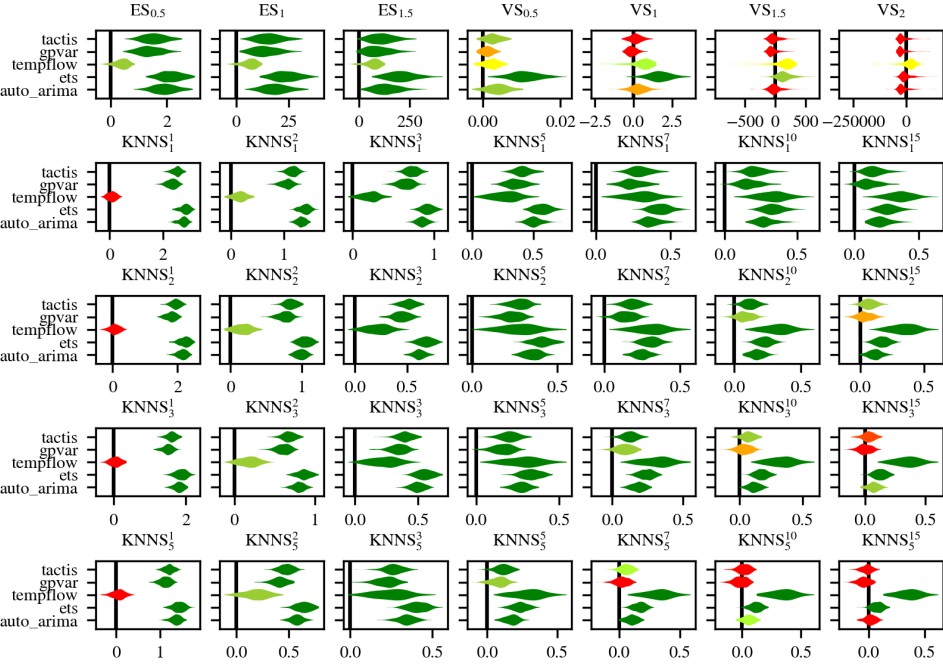

Figure 3: FRED-MD Dataset with timegrad as the data generating model.

We regard each of these models as the data generating process $p_0$ for each dataset. For each such pair, we synthesize future observations $\boldsymbol{y}$ from each historic sample from the test split of the respective dataset. We use the normalization scheme used in the GluonTS framework (Alexandrov et al., 2020) before evaluating all metrics: a single normalization factor is computed from the test dataset by averaging the absolute value of the observed future of all time series and time steps.

As the VS is computationally costly, we only evaluate it for the smallest dataset used by Drouin et al. (2022), which is FRED-MD (Godahewa et al., 2021). This dataset also poses the biggest issues for our KNNS Score, especially if timegrad (Rasul et al., 2021a) is the data generating process. Figure 3 shows violin plots collected over 200 repetitions, see Appendix C for details and further results. A clear trend is visible: for $k = 1$ the KNNS Score has difficulties discriminating between timegrad and tempflow (Rasul et al., 2021b). If $k$ and $d$ are large, gpvar (Salinas et al., 2019), tactis (Drouin et al., 2022) and auto_arima become indistinguishable from timegrad. In between, KNNS produces correct evaluations. ES also provides good results, whereas VS fails.

## 6  LIMITATIONS AND CONCLUSION

As univariate evaluations fail to measure dependencies between different time steps and different time series, multivariate evaluation is crucial. We prove that the Energy Score is equivalent to measuring the average CRPS score over random projections. This leads to an information loss which makes the Energy Score fail to gather correlation structure. The Variogram Score is not rotation invariant, such that the direction of a deviation between observation and prediction changes the result which is not desired in general settings. In addition, it is rather costly to compute.

We propose the KNNS Score, a novel metric for probabilistic multivariate forecasting, which is based on $k$-nearest neighbor density estimation. Density estimation in a very high dimensional space would require an impossible amount of samples, thus we perform random projections into low-dimensional space and compute an average distance to the $k$-nearest neighbor.

One drawback of this metric is the need for numerous random projections. Investigating a closed-form solution is a possibility for future research. Additionally, the choice of parameters, namely $k$ and $d$, impacts the outcomes. Therefore, establishing guidelines for selecting these parameters during evaluation will be an important topic for future work. Through a series of experiments, we demonstrate that the KNNS Score effectively discriminates the true data generating process from various alternative models, particularly in the context of contemporary deep learning models.

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

# A PROOF FOR THEOREM 1

**Theorem 1.** *Let $r$ be a $d$-dimensional random vector with unit length. Then there exists a constant $c_d$, such that*

$$\mathbb{E}_r \text{CRPS}(\{r \cdot x : x \in \mathbb{X}\}, r \cdot y) = c_d \cdot \text{ES}_1(\mathbb{X}, y). \tag{10}$$

*Proof.*

$$\mathbb{E}_r \text{CRPS}(\{r \cdot x : x \in \mathbb{X}\}, r \cdot y) \tag{15}$$

$$= \frac{1}{2} \mathbb{E}_{\substack{x,x' \in \mathbb{X} \\ x \neq x'}} \mathbb{E}_r |r \cdot (x - x')| - \mathbb{E}_{x \in \mathbb{X}} \mathbb{E}_r |r \cdot (x - y)| \tag{16}$$

$$= \frac{1}{2} \mathbb{E}_{\substack{x,x' \in \mathbb{X} \\ x \neq x'}} \mathbb{E}_r \|r\| \|x - x'\| |\cos(\theta_{r,x-x'})| - \mathbb{E}_{x \in \mathbb{X}} \mathbb{E}_r \|r\| \|x - y\| |\cos(\theta_{r,x-y})| \tag{17}$$

$$= \frac{1}{2} \mathbb{E}_{\substack{x,x' \in \mathbb{X} \\ x \neq x'}} \mathbb{E}_r \|x - x'\| |\cos \theta_{r,e_1}| - \mathbb{E}_{x \in \mathbb{X}} \mathbb{E}_r \|x - y\| |\cos \theta_{r,e_1}| \tag{18}$$

$$= \mathbb{E}_r |\cos \theta_{r,e_1}| \cdot \left( \frac{1}{2} \mathbb{E}_{\substack{x,x' \in \mathbb{X} \\ x \neq x'}} \|x - x'\| - \mathbb{E}_{x \in \mathbb{X}} \|x - y\| \right) \tag{19}$$

$$= c_d \cdot \text{ES}_1(\mathbb{X}, y) \tag{20}$$

Here, $\theta_{a,b}$ denotes the angle between $a$ and $b$. Since $r$ is a random vector with unit length, $\mathbb{E}_r |\cos(\theta_{r,b})| = \mathbb{E}_r |\cos(\theta_{r,e_1})|$ where $e_1$ is the base vector $[1, 0, \ldots, 0]^\top$. The dimension-dependent constant $c_d$ can be computed as

$$c_d = \mathbb{E}_r |\cos \theta_{r,e_1}| = \frac{1}{\sqrt{\pi}} \frac{\Gamma\left(\frac{d}{2}\right)}{\Gamma\left(\frac{d+1}{2}\right)}, \tag{21}$$

where $\Gamma$ denotes the gamma function. See Lemma 1 for the derivation. $\square$

**Lemma 1.** *Let $r$ be a random vector sampled uniformly from the surface of a $d$-dimensional unit sphere and let $e_1 = [1, 0, \ldots, 0]^\top$. The expected cosine between these vectors can be computed as*

$$\mathbb{E}_r |\cos \theta_{r,e_1}| = \frac{1}{\sqrt{\pi}} \frac{\Gamma\left(\frac{d}{2}\right)}{\Gamma\left(\frac{d+1}{2}\right)}. \tag{22}$$

*Proof.* If $\theta$ is the angle between two arbitrary random vectors, then $X = \sin^2 \theta$ has the probability density function

$$\rho(x) = \frac{1}{B\left(\frac{d-1}{2}, \frac{1}{2}\right)} \frac{x^{\frac{d-1}{2} - 1}}{\sqrt{1 - x}}, \tag{23}$$

where $B(\alpha, \beta)$ is the normalization factor of the beta distribution (Frankl & Maehara, 1990). Normalization of these vectors to unit length does not change the angles and fixing one of the vectors to $e_1$ and only sampling the other one does not change the distribution. Thus, using $\sin^2 \theta + \cos^2 \theta = 1$, we can compute the expected absolute cosine as

$$\mathbb{E}_r |\cos \theta_{r,e_1}| = \int_0^1 \sqrt{1 - x} \rho(x) \mathrm{d}x \tag{24}$$

$$= \frac{1}{B\left(\frac{d-1}{2}, \frac{1}{2}\right)} \int_0^1 x^{\frac{d-1}{2} - 1} \mathrm{d}x \tag{25}$$

$$= \frac{\Gamma\left(\frac{d}{2}\right)}{\Gamma\left(\frac{1}{2}\right) \Gamma\left(\frac{d-1}{2}\right)} \frac{1}{\frac{d-1}{2}} \tag{26}$$

$$= \frac{1}{\sqrt{\pi}} \frac{\Gamma\left(\frac{d}{2}\right)}{\Gamma\left(\frac{d+1}{2}\right)} \tag{27}$$

where we applied the definition of the Beta normalization factor followed by the recurrence relation of the Gamma function. $\square$

# B   FURTHER RESULTS ON SYNTHETIC DATA

This section shows the results for all experiments we conducted on synthetic data. If a metric can discriminate between the true data generating process and the adapted models, $DS_c$ and $DS_a$ are close to $0$. Values close to $1$ mean that the metric scores the wrong model better.

| | $DS_c$ | | | | | | |
|---|---|---|---|---|---|---|---|
| | mean bias | larger var | smaller var | less corr | more corr | corr (i) | corr (ii) |
| $ES_{0.5}$ | 0.00 | 0.00 | 0.00 | 0.40 | 0.34 | 0.04 | 0.42 |
| $ES_1$ | 0.00 | 0.00 | 0.00 | 0.42 | 0.44 | 0.10 | 0.42 |
| $ES_{1.5}$ | 0.00 | 0.00 | 0.00 | 0.50 | 0.52 | 0.26 | 0.46 |
| $VS_{0.5}$ | 0.14 | 0.00 | 0.00 | 0.00 | 0.00 | 0.00 | 0.00 |
| $VS_1$ | 0.10 | 0.00 | 0.00 | 0.00 | 0.00 | 0.00 | 0.00 |
| $VS_{1.5}$ | 0.08 | 0.00 | 0.00 | 0.02 | 0.02 | 0.00 | 0.06 |
| $VS_2$ | 0.08 | 0.00 | 0.00 | 0.02 | 0.12 | 0.00 | 0.06 |
| $KNN_1$ | 0.00 | 0.00 | 1.00 | 0.00 | 1.00 | 1.00 | 0.00 |
| $KNNS_1^1$ | 0.00 | 1.00 | 0.00 | 0.60 | 0.00 | 0.00 | 0.36 |
| $KNNS_2^1$ | 0.00 | 1.00 | 0.00 | 0.62 | 0.00 | 0.00 | 0.40 |
| $KNNS_3^1$ | 0.00 | 1.00 | 0.00 | 0.64 | 0.02 | 0.00 | 0.46 |
| $KNNS_4^1$ | 0.00 | 1.00 | 0.00 | 0.70 | 0.02 | 0.00 | 0.52 |
| $KNNS_5^1$ | 0.00 | 1.00 | 0.00 | 0.68 | 0.06 | 0.00 | 0.54 |
| $KNNS_1^2$ | 0.00 | 1.00 | 0.00 | 0.30 | 0.00 | 0.00 | 0.10 |
| $KNNS_2^2$ | 0.00 | 0.00 | 0.00 | 0.16 | 0.10 | 0.00 | 0.08 |
| $KNNS_3^2$ | 0.00 | 0.00 | 0.00 | 0.06 | 0.18 | 0.00 | 0.06 |
| $KNNS_4^2$ | 0.00 | 0.00 | 0.00 | 0.02 | 0.32 | 0.00 | 0.04 |
| $KNNS_5^2$ | 0.00 | 0.00 | 0.00 | 0.02 | 0.46 | 0.00 | 0.04 |
| $KNNS_1^3$ | 0.00 | 0.00 | 0.00 | 0.00 | 0.26 | 0.00 | 0.00 |
| $KNNS_2^3$ | 0.00 | 0.00 | 0.00 | 0.00 | 0.82 | 0.00 | 0.00 |
| $KNNS_3^3$ | 0.00 | 0.00 | 0.16 | 0.00 | 1.00 | 0.10 | 0.00 |
| $KNNS_4^3$ | 0.00 | 0.00 | 1.00 | 0.00 | 1.00 | 0.40 | 0.00 |
| $KNNS_5^3$ | 0.00 | 0.00 | 1.00 | 0.00 | 1.00 | 0.74 | 0.00 |
| $KNNS_1^4$ | 0.00 | 0.00 | 0.00 | 0.00 | 0.92 | 0.00 | 0.00 |
| $KNNS_2^4$ | 0.00 | 0.00 | 1.00 | 0.00 | 1.00 | 0.42 | 0.00 |
| $KNNS_3^4$ | 0.00 | 0.00 | 1.00 | 0.00 | 1.00 | 0.90 | 0.00 |
| $KNNS_4^4$ | 0.00 | 0.00 | 1.00 | 0.00 | 1.00 | 0.98 | 0.00 |
| $KNNS_5^4$ | 0.00 | 0.00 | 1.00 | 0.00 | 1.00 | 1.00 | 0.00 |
| $KNNS_1^5$ | 0.00 | 0.00 | 1.00 | 0.00 | 1.00 | 0.40 | 0.00 |
| $KNNS_2^5$ | 0.00 | 0.00 | 1.00 | 0.00 | 1.00 | 1.00 | 0.00 |
| $KNNS_3^5$ | 0.00 | 0.00 | 1.00 | 0.00 | 1.00 | 1.00 | 0.00 |
| $KNNS_4^5$ | 0.00 | 0.00 | 1.00 | 0.00 | 1.00 | 1.00 | 0.00 |
| $KNNS_5^5$ | 0.00 | 0.00 | 1.00 | 0.00 | 1.00 | 1.00 | 0.00 |

| | $DS_a$ | | | | | | |
|---|---|---|---|---|---|---|---|
| | mean bias | larger var | smaller var | less corr | more corr | corr (i) | corr (ii) |
| $ES_{0.5}$ | 0.00 | 0.00 | 0.00 | 0.34 | 0.18 | 0.02 | 0.29 |
| $ES_1$ | 0.00 | 0.00 | 0.00 | 0.43 | 0.29 | 0.05 | 0.37 |
| $ES_{1.5}$ | 0.00 | 0.00 | 0.00 | 0.53 | 0.42 | 0.22 | 0.47 |
| $VS_{0.5}$ | 0.06 | 0.00 | 0.00 | 0.00 | 0.00 | 0.00 | 0.00 |
| $VS_1$ | 0.05 | 0.00 | 0.00 | 0.00 | 0.00 | 0.00 | 0.00 |
| $VS_{1.5}$ | 0.04 | 0.00 | 0.00 | 0.00 | 0.01 | 0.00 | 0.01 |
| $VS_2$ | 0.03 | 0.00 | 0.00 | 0.01 | 0.04 | 0.00 | 0.04 |
| $KNN_1$ | 0.00 | 0.00 | 1.00 | 0.00 | 1.00 | 1.00 | 0.00 |
| $KNNS_1^1$ | 0.00 | 1.00 | 0.00 | 0.66 | 0.00 | 0.00 | 0.31 |
| $KNNS_2^1$ | 0.00 | 1.00 | 0.00 | 0.69 | 0.00 | 0.00 | 0.36 |
| $KNNS_3^1$ | 0.00 | 1.00 | 0.00 | 0.72 | 0.00 | 0.00 | 0.41 |
| $KNNS_4^1$ | 0.00 | 1.00 | 0.00 | 0.74 | 0.00 | 0.00 | 0.44 |
| $KNNS_5^1$ | 0.00 | 1.00 | 0.00 | 0.73 | 0.00 | 0.00 | 0.46 |
| $KNNS_1^2$ | 0.00 | 1.00 | 0.00 | 0.11 | 0.00 | 0.00 | 0.04 |
| $KNNS_2^2$ | 0.00 | 0.00 | 0.00 | 0.04 | 0.01 | 0.00 | 0.04 |
| $KNNS_3^2$ | 0.00 | 0.00 | 0.00 | 0.01 | 0.06 | 0.00 | 0.03 |
| $KNNS_4^2$ | 0.00 | 0.00 | 0.00 | 0.00 | 0.23 | 0.00 | 0.02 |
| $KNNS_5^2$ | 0.00 | 0.00 | 0.00 | 0.00 | 0.46 | 0.00 | 0.02 |
| $KNNS_1^3$ | 0.00 | 0.00 | 0.00 | 0.00 | 0.19 | 0.00 | 0.00 |
| $KNNS_2^3$ | 0.00 | 0.00 | 0.00 | 0.00 | 0.94 | 0.00 | 0.00 |
| $KNNS_3^3$ | 0.00 | 0.00 | 0.03 | 0.00 | 1.00 | 0.01 | 0.00 |
| $KNNS_4^3$ | 0.00 | 0.00 | 1.00 | 0.00 | 1.00 | 0.32 | 0.00 |
| $KNNS_5^3$ | 0.00 | 0.00 | 1.00 | 0.00 | 1.00 | 0.79 | 0.00 |
| $KNNS_1^4$ | 0.00 | 0.00 | 0.00 | 0.00 | 0.98 | 0.00 | 0.00 |
| $KNNS_2^4$ | 0.00 | 0.00 | 1.00 | 0.00 | 1.00 | 0.35 | 0.00 |
| $KNNS_3^4$ | 0.00 | 0.00 | 1.00 | 0.00 | 1.00 | 0.96 | 0.00 |
| $KNNS_4^4$ | 0.00 | 0.00 | 1.00 | 0.00 | 1.00 | 1.00 | 0.00 |
| $KNNS_5^4$ | 0.00 | 0.00 | 1.00 | 0.00 | 1.00 | 1.00 | 0.00 |
| $KNNS_1^5$ | 0.00 | 0.00 | 1.00 | 0.00 | 1.00 | 0.36 | 0.00 |
| $KNNS_2^5$ | 0.00 | 0.00 | 1.00 | 0.00 | 1.00 | 1.00 | 0.00 |
| $KNNS_3^5$ | 0.00 | 0.00 | 1.00 | 0.00 | 1.00 | 1.00 | 0.00 |
| $KNNS_4^5$ | 0.00 | 0.00 | 1.00 | 0.00 | 1.00 | 1.00 | 0.00 |
| $KNNS_5^5$ | 0.00 | 0.00 | 1.00 | 0.00 | 1.00 | 1.00 | 0.00 |

| | DS$_c$ | | | | DS$_a$ | | | |
|---|---|---|---|---|---|---|---|---|
| | mean distance | | weights | | mean distance | | weights | |
| | higher | lower | similar | different | higher | lower | similar | different |
| ES$_{0.5}$ | 0.32 | 0.38 | 0.50 | 0.34 | 0.16 | 0.27 | 0.44 | 0.37 |
| ES$_1$ | 0.32 | 0.42 | 0.52 | 0.36 | 0.17 | 0.28 | 0.44 | 0.37 |
| ES$_{1.5}$ | 0.32 | 0.44 | 0.52 | 0.36 | 0.17 | 0.28 | 0.44 | 0.38 |
| VS$_{0.5}$ | 0.04 | 0.50 | 0.48 | 0.42 | 0.00 | 0.52 | 0.51 | 0.33 |
| VS$_1$ | 0.02 | 0.54 | 0.52 | 0.42 | 0.00 | 0.53 | 0.55 | 0.33 |
| VS$_{1.5}$ | 0.02 | 0.60 | 0.52 | 0.38 | 0.00 | 0.54 | 0.57 | 0.33 |
| VS$_2$ | 0.02 | 0.60 | 0.58 | 0.36 | 0.00 | 0.54 | 0.58 | 0.33 |
| KNN$_1$ | 0.22 | 0.58 | 0.50 | 0.44 | 0.20 | 0.58 | 0.56 | 0.54 |
| KNNS$_1^1$ | 0.70 | 0.52 | 0.56 | 0.48 | 0.81 | 0.47 | 0.61 | 0.46 |
| KNNS$_2^1$ | 0.68 | 0.46 | 0.56 | 0.44 | 0.77 | 0.39 | 0.58 | 0.37 |
| KNNS$_3^1$ | 0.68 | 0.40 | 0.46 | 0.44 | 0.75 | 0.39 | 0.57 | 0.37 |
| KNNS$_4^1$ | 0.68 | 0.42 | 0.48 | 0.46 | 0.76 | 0.39 | 0.55 | 0.37 |
| KNNS$_5^1$ | 0.68 | 0.40 | 0.52 | 0.44 | 0.76 | 0.38 | 0.55 | 0.37 |
| KNNS$_1^2$ | 0.46 | 0.44 | 0.50 | 0.52 | 0.45 | 0.47 | 0.57 | 0.46 |
| KNNS$_2^2$ | 0.36 | 0.48 | 0.48 | 0.40 | 0.31 | 0.39 | 0.53 | 0.39 |
| KNNS$_3^2$ | 0.32 | 0.42 | 0.44 | 0.36 | 0.24 | 0.34 | 0.49 | 0.38 |
| KNNS$_4^2$ | 0.28 | 0.46 | 0.50 | 0.36 | 0.20 | 0.35 | 0.48 | 0.38 |
| KNNS$_5^2$ | 0.26 | 0.46 | 0.48 | 0.34 | 0.17 | 0.35 | 0.46 | 0.38 |
| KNNS$_1^3$ | 0.34 | 0.40 | 0.46 | 0.50 | 0.22 | 0.40 | 0.52 | 0.56 |
| KNNS$_2^3$ | 0.26 | 0.42 | 0.42 | 0.48 | 0.15 | 0.41 | 0.48 | 0.52 |
| KNNS$_3^3$ | 0.20 | 0.46 | 0.42 | 0.50 | 0.09 | 0.42 | 0.51 | 0.52 |
| KNNS$_4^3$ | 0.18 | 0.44 | 0.40 | 0.46 | 0.06 | 0.41 | 0.49 | 0.48 |
| KNNS$_5^3$ | 0.16 | 0.44 | 0.42 | 0.48 | 0.05 | 0.42 | 0.49 | 0.49 |
| KNNS$_1^4$ | 0.18 | 0.46 | 0.46 | 0.52 | 0.13 | 0.43 | 0.53 | 0.43 |
| KNNS$_2^4$ | 0.14 | 0.48 | 0.54 | 0.48 | 0.09 | 0.46 | 0.55 | 0.45 |
| KNNS$_3^4$ | 0.12 | 0.48 | 0.56 | 0.44 | 0.07 | 0.48 | 0.52 | 0.42 |
| KNNS$_4^4$ | 0.10 | 0.48 | 0.56 | 0.46 | 0.05 | 0.47 | 0.52 | 0.40 |
| KNNS$_5^4$ | 0.08 | 0.48 | 0.56 | 0.46 | 0.04 | 0.48 | 0.52 | 0.42 |
| KNNS$_1^5$ | 0.12 | 0.46 | 0.46 | 0.44 | 0.08 | 0.38 | 0.44 | 0.41 |
| KNNS$_2^5$ | 0.08 | 0.46 | 0.50 | 0.44 | 0.06 | 0.41 | 0.47 | 0.46 |
| KNNS$_3^5$ | 0.10 | 0.50 | 0.48 | 0.48 | 0.05 | 0.44 | 0.48 | 0.47 |
| KNNS$_4^5$ | 0.10 | 0.48 | 0.48 | 0.48 | 0.05 | 0.45 | 0.48 | 0.50 |
| KNNS$_5^5$ | 0.06 | 0.50 | 0.50 | 0.46 | 0.04 | 0.47 | 0.48 | 0.50 |

| | $DS_c$ | | | | $DS_a$ | | | |
| --- | --- | --- | --- | --- | --- | --- | --- | --- |
| | mean distance | | weights | | mean distance | | weights | |
| | higher | lower | similar | different | higher | lower | similar | different |
| $ES_{0.5}$ | 0.34 | 0.46 | 0.46 | 0.52 | 0.27 | 0.48 | 0.46 | 0.53 |
| $ES_1$ | 0.34 | 0.44 | 0.44 | 0.52 | 0.27 | 0.48 | 0.45 | 0.53 |
| $ES_{1.5}$ | 0.36 | 0.44 | 0.44 | 0.54 | 0.26 | 0.48 | 0.45 | 0.52 |
| $VS_{0.5}$ | 0.14 | 0.52 | 0.56 | 0.44 | 0.04 | 0.52 | 0.57 | 0.45 |
| $VS_1$ | 0.10 | 0.54 | 0.56 | 0.38 | 0.02 | 0.54 | 0.57 | 0.43 |
| $VS_{1.5}$ | 0.04 | 0.52 | 0.56 | 0.42 | 0.01 | 0.56 | 0.57 | 0.42 |
| $VS_2$ | 0.04 | 0.54 | 0.58 | 0.40 | 0.01 | 0.58 | 0.56 | 0.40 |
| $KNN_1$ | 0.04 | 0.70 | 0.54 | 0.54 | 0.01 | 0.74 | 0.57 | 0.45 |
| $KNNS_1^1$ | 0.98 | 0.28 | 0.48 | 0.52 | 1.00 | 0.25 | 0.42 | 0.58 |
| $KNNS_2^1$ | 0.98 | 0.34 | 0.50 | 0.54 | 0.99 | 0.18 | 0.42 | 0.58 |
| $KNNS_3^1$ | 0.98 | 0.30 | 0.46 | 0.60 | 0.99 | 0.17 | 0.44 | 0.62 |
| $KNNS_4^1$ | 0.92 | 0.32 | 0.44 | 0.56 | 0.98 | 0.19 | 0.43 | 0.58 |
| $KNNS_5^1$ | 0.90 | 0.30 | 0.42 | 0.52 | 0.96 | 0.20 | 0.42 | 0.57 |
| $KNNS_1^2$ | 0.78 | 0.42 | 0.42 | 0.50 | 0.86 | 0.35 | 0.40 | 0.51 |
| $KNNS_2^2$ | 0.58 | 0.38 | 0.46 | 0.42 | 0.70 | 0.38 | 0.40 | 0.44 |
| $KNNS_3^2$ | 0.46 | 0.38 | 0.46 | 0.44 | 0.53 | 0.41 | 0.41 | 0.45 |
| $KNNS_4^2$ | 0.30 | 0.40 | 0.42 | 0.46 | 0.38 | 0.42 | 0.41 | 0.46 |
| $KNNS_5^2$ | 0.28 | 0.40 | 0.44 | 0.46 | 0.27 | 0.47 | 0.43 | 0.46 |
| $KNNS_1^3$ | 0.40 | 0.48 | 0.46 | 0.40 | 0.33 | 0.52 | 0.40 | 0.37 |
| $KNNS_2^3$ | 0.20 | 0.54 | 0.38 | 0.42 | 0.09 | 0.62 | 0.42 | 0.36 |
| $KNNS_3^3$ | 0.08 | 0.60 | 0.38 | 0.42 | 0.01 | 0.64 | 0.41 | 0.36 |
| $KNNS_4^3$ | 0.04 | 0.60 | 0.42 | 0.44 | 0.00 | 0.71 | 0.45 | 0.36 |
| $KNNS_5^3$ | 0.04 | 0.64 | 0.42 | 0.40 | 0.00 | 0.75 | 0.47 | 0.36 |
| $KNNS_1^4$ | 0.16 | 0.50 | 0.40 | 0.52 | 0.12 | 0.61 | 0.41 | 0.45 |
| $KNNS_2^4$ | 0.10 | 0.54 | 0.46 | 0.48 | 0.03 | 0.70 | 0.46 | 0.43 |
| $KNNS_3^4$ | 0.02 | 0.60 | 0.42 | 0.48 | 0.01 | 0.74 | 0.45 | 0.42 |
| $KNNS_4^4$ | 0.02 | 0.62 | 0.42 | 0.48 | 0.00 | 0.76 | 0.45 | 0.40 |
| $KNNS_5^4$ | 0.02 | 0.62 | 0.42 | 0.48 | 0.00 | 0.80 | 0.45 | 0.40 |
| $KNNS_1^5$ | 0.06 | 0.60 | 0.38 | 0.44 | 0.01 | 0.62 | 0.46 | 0.41 |
| $KNNS_2^5$ | 0.00 | 0.62 | 0.44 | 0.40 | 0.00 | 0.71 | 0.50 | 0.41 |
| $KNNS_3^5$ | 0.00 | 0.64 | 0.50 | 0.40 | 0.00 | 0.75 | 0.51 | 0.37 |
| $KNNS_4^5$ | 0.00 | 0.66 | 0.52 | 0.42 | 0.00 | 0.77 | 0.50 | 0.35 |
| $KNNS_5^5$ | 0.00 | 0.66 | 0.54 | 0.40 | 0.00 | 0.80 | 0.52 | 0.35 |

## C   FURTHER RESULTS ON REAL DATA

The violin plots presented in the following figures visualize the distribution of differences $S(\boldsymbol{y}, \mathbb{X}_0) - S(\boldsymbol{y}, \mathbb{X}_1)$ for different data generating models $p_0$ on the FRED-MD dataset. Ideally, this difference is positive, because forecasts of the data generating model should score better than forecasts of other models. A metric demonstrates strong discrimination ability when the majority of the probability mass resides on the positive side of the zero-line. The color of each violin plot visualizes $DS_c$, i.e. the proportion of points lying on the negative side.

Apart from discrimination ability, we can also see that the metrics produce different rankings of the models. For example in Figure 7, the Energy Score rates auto_arima to be the closest to tactis, followed by gpvar and ets. For the Variogram Score, the order depends on $p$, for small $p$, ets is closer to tactis, for larger $p$ the ordering is the same as for the Energy Score. Our KNN Score ranks gpvar first, if $k = 1, d = 2$. For bigger $k, d$ auto_arima is ranked best.

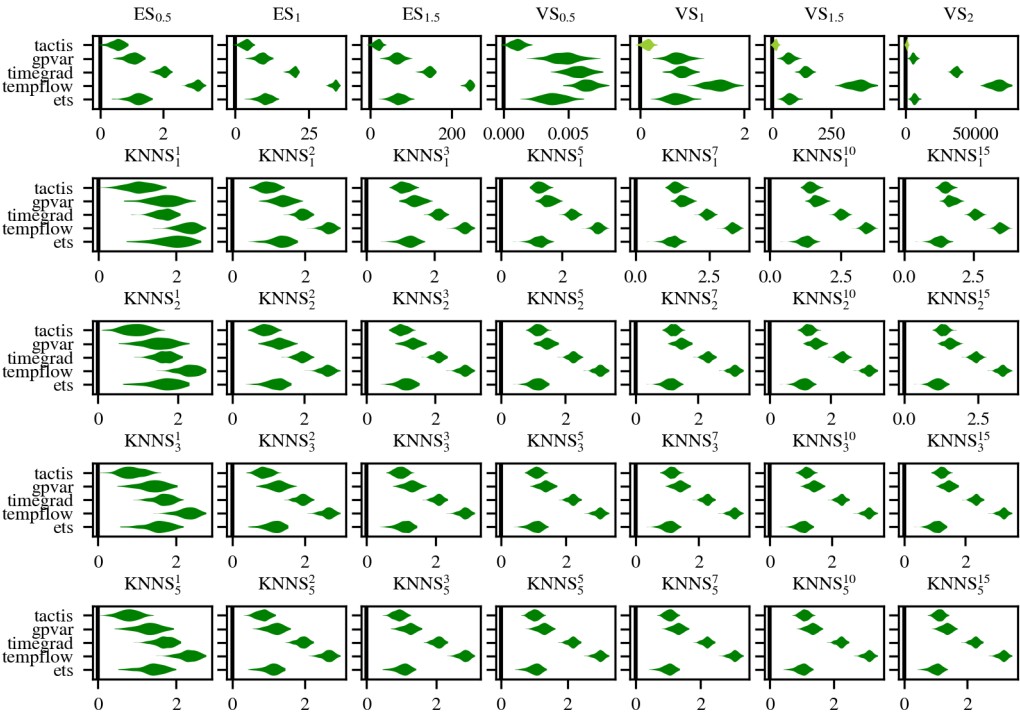

Figure 4: FRED-MD Dataset, data generating model is auto_arima

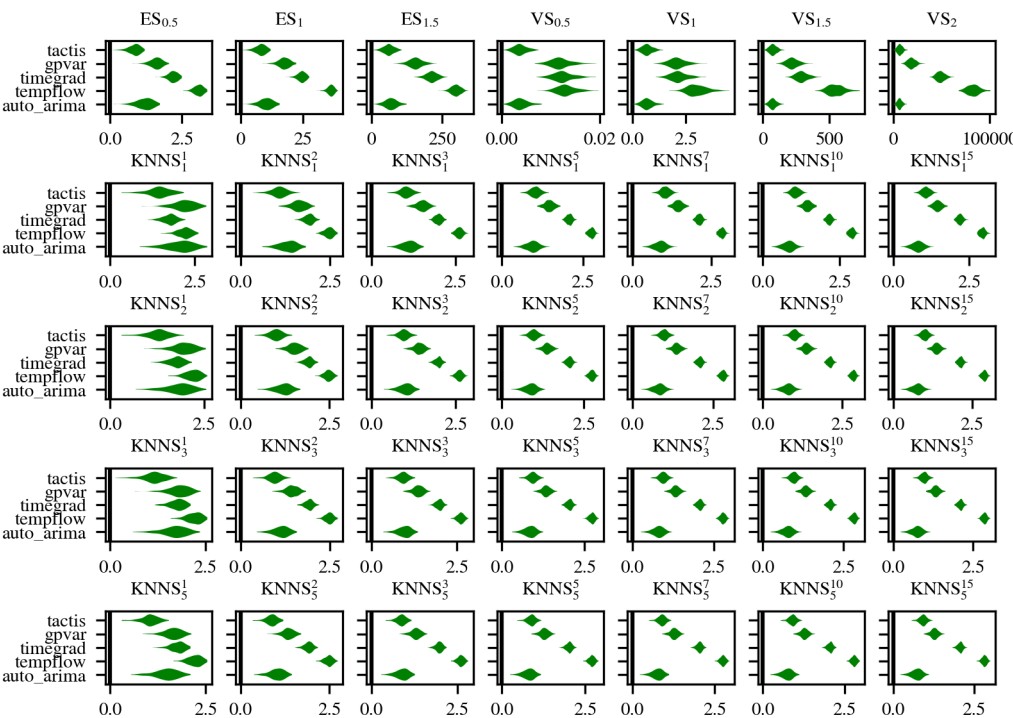

Figure 5: FRED-MD Dataset, data generating model is ets

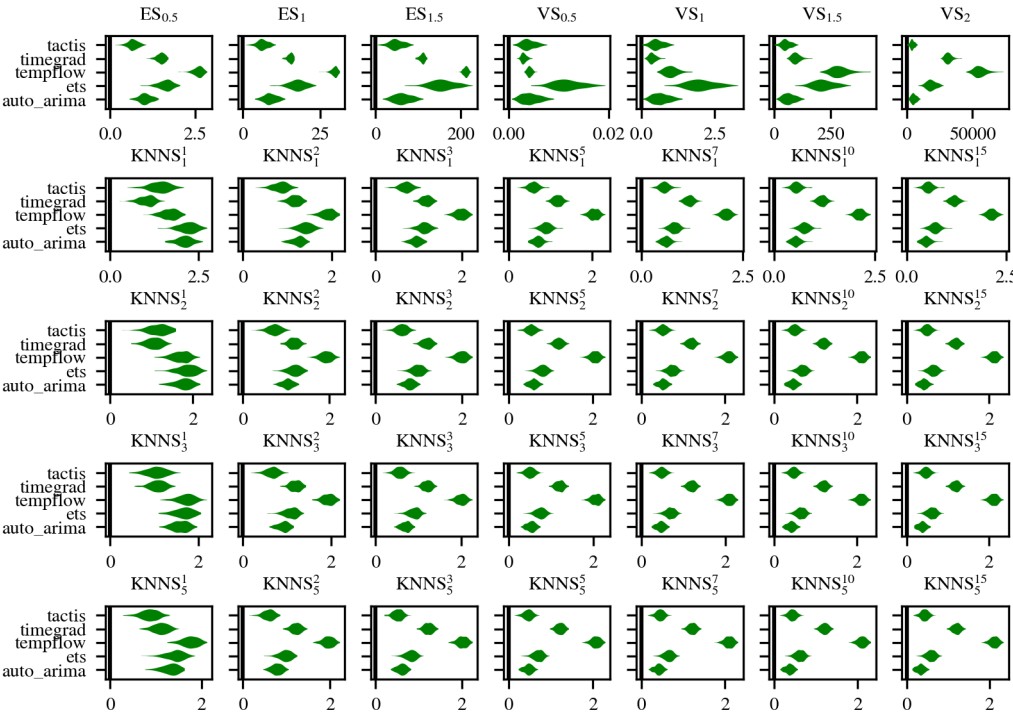

Figure 6: FRED-MD Dataset, data generating model is gpvar

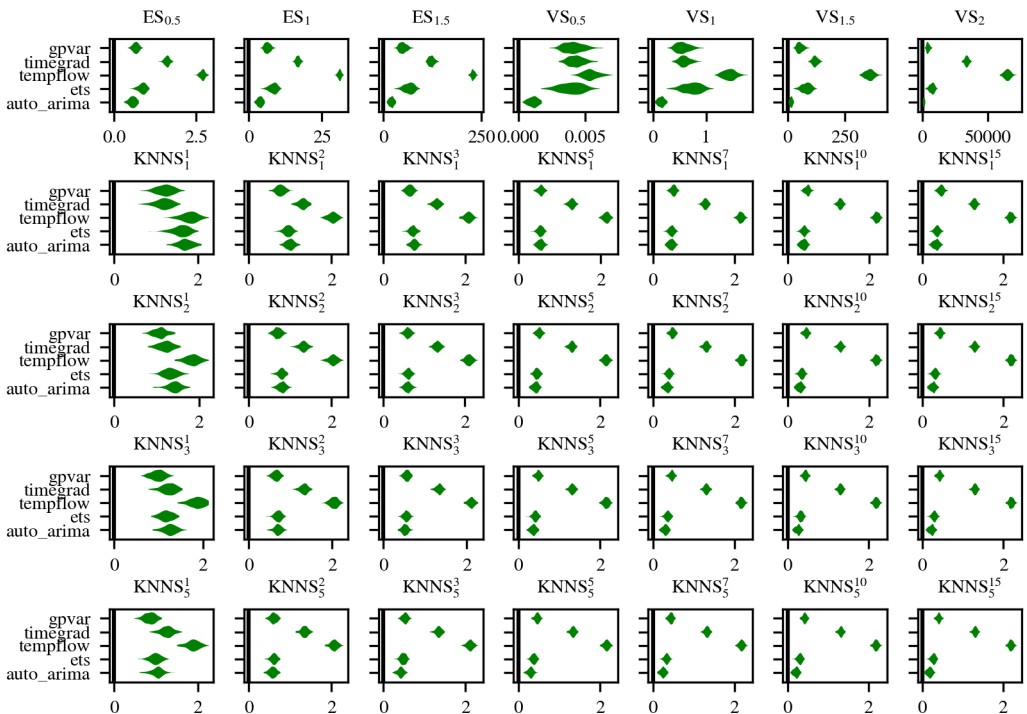

Figure 7: FRED-MD Dataset, data generating model is tactis

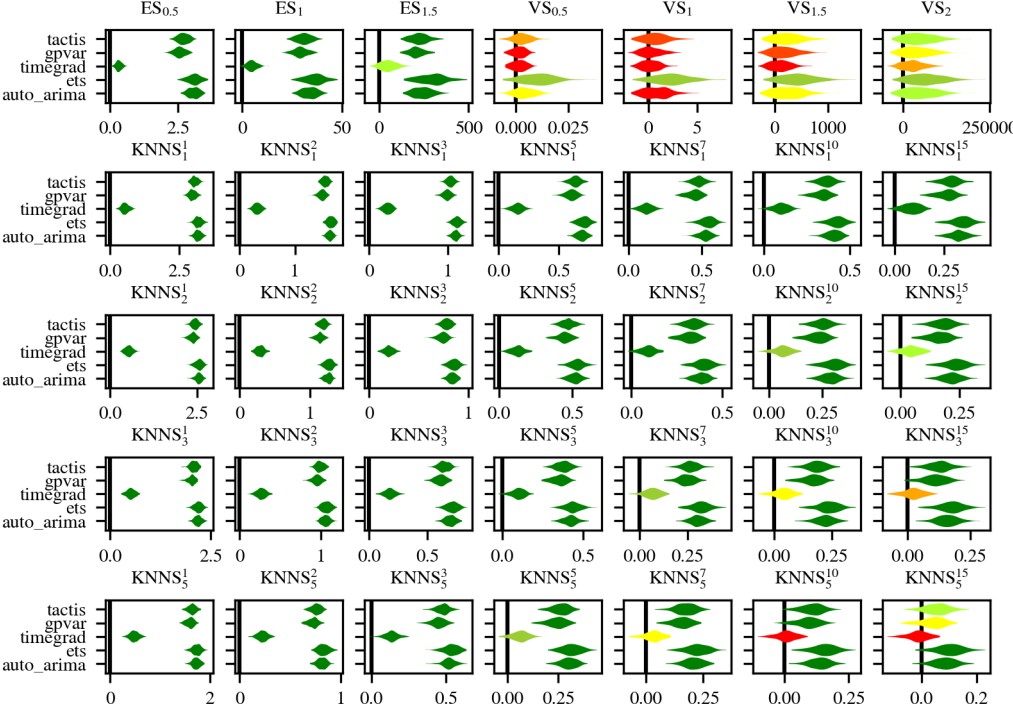

Figure 8: FRED-MD Dataset, data generating model is tempflow

