# OpenReview forum: "The KNN Score for Evaluating Probabilistic Multivariate Time Series Forecasting"
_ICLR.cc/2024/Conference — Submitted to ICLR 2024_

### Official Review · Reviewer_hvQm · 2023-10-29

**Soundness:** 2 fair
**Presentation:** 3 good
**Contribution:** 1 poor
**Rating:** 3
**Confidence:** 4

**Summary:**

The paper proposes the KNN score for evaluating multivariate time series forecasts. The paper first gives a broad overview of multivariate distributional forecasting metrics and identifies their failure modes. It then proposes the KNNS metric, motivated by the fact that out-of-sample likelihood would be a good score. The KNNS metric is then compared to baselines on synthetic and small-sized real world scenarios.

However, the proposal appears to move from an incorrect premise. The metric itself appears rather complicated, difficult to implement in practice, and requires the correct tuning of hyperparameters--the choices of which may be data and model dependent. The experimental setup is not nearly enough to substantiate the introduction of a new forecast evaluation metric, and the results fail to demonstrate conclusive evidence.

**Strengths:**

The paper is well-written overall and adresses an important problem. It gives a broad overview and critique of previous evaluation methods used in other papers and identifies their failure modes. After introducing KNNS, it demonstrates an interesting link between random projections, energy scores and CRPS. The authors then use this insight to make the otherwise very compute-intensive KNNS metric somewhat manageable (computationally, albeit not statistically) in practice.

The paper also offers good insight about how the statistical power of a forecast evaluation metric was evaluated in prior literature, and proposes a new and thoughtful experiment setup. The authors are clear in their conclusions and about the limitations of their work.

**Weaknesses:**

Firstly, the paper moves from the premise that a metric used for model selection in forecasting must be able to correctly capture a variety of properties about forecasts including how much the forecasts exhibit correlation and temporal regularity. However, I would argue this is only true to the extent that it makes the metric more "sample efficient" with higher statistical power.

While simple MAD, MSE, CRPS may be oblivious to correlations or regularity of forecasts, they reflect the key desiderata of the forecasts stemming from the task: that they are close to the ground truth. As the model is better able to capture correlations, one expects that simple metrics like MSE will also vanish. In other terms, one does not require metrics are highly representative---but that they are consistent and proper. For added complexity of metrics, one should argue that the metrics result in higher statistical power under realistic finite sample constraints and forecasting scenarios (realistic true distributions). Given the inconclusivity of the paper's empirical findings and the lack of a framework for the tuning of KNNS parameter I do not believe this bar has been cleared for KNNS.

My second critique would be on the metric itself. Motivated by kNN density estimation, the authors introduce the L2 distance to the kth nearest neighbor as the model selection metric, despite high dimensionality. Besides the fact that this could result in notoriously high sampling variance depending on the true data distribution, this makes the magnitude of the metric dependent on the size of the sample (in the paper, "ensemble"). i.e., in order to compare two models one would have to compare them on the same number of sampled trajectories or the metric would be invalid otherwise.

Finally, the KNN score's intuition is that the best model is able to place a forecast close in L2 distance to the ground truth. Note that for a univariate forecast this is equivalent to saying that the k-th best forecast in an ensemble, measured in squared error, has low squared error. Setting k=1, this is equivalent to choosing the model with the "best in hindsight" MSE. In other words, KNNS does not appear to measure the quality of distributional forecasts, a desired property set out in the paper, but only point forecast error.

Some other points

- The introduction of the random projections could be better motivated. Multiple random projections sampled and with sufficient latent dimension would be justified for only very high dimensionality tasks such as spatiotemporal forecasting in earth sciences, etc. This doesn't appear to be the case with the experiment setup.
- The introduction of KNNS is somewhat counterintuitive. The paper first reports that out of sample likelihood (i.e., perplexity) is a desirable metric; but due to its practicality immediately moves to KNN distance. I believe this link should be substantiated.
- $d$ is redefined in the paper as 'difference' although interchangeably also used to denote dimensions. The notations $\mathbb{y}$ and $\mathbb{Y}$ (resp. X) also appear to have differing definitions through the paper.

**Questions:**

Why didn't you consider using the average of the first k distances to decrease variance?

---

> ### Author Response · Authors · 2023-11-20
>
> Thank you for your feedback! We see from the reviews that the paper needs significant revision and would like to use this forum as an opportunity for further clarification and feedback.
>
> **“While simple MAD, MSE, CRPS may be oblivious to correlations or regularity of forecasts, they reflect the key desiderata of the forecasts stemming from the task: that they are close to the ground truth. As the model is better able to capture correlations, one expects that simple metrics like MSE will also vanish.”**
>
> A probabilistic prediction ideally represents the uncertainty of the true data distribution correctly. Assume a distribution with two modes of equal size, then the MSE would be ideal if the model always predicts the mean of the distribution, where the density could be zero. A probabilistic univariate score would indeed vanish, e.g. an optimal probabilistic multivariate model would receive the optimal CRPS score, but the reverse is not true: a model that is optimal under CRPS does not have to produce the correct multivariate structure. Therefore, more complex scoring rules are needed to properly evaluate multivariate time series forecasts.
>
> **“this makes the magnitude of the metric dependent on the size of the sample (in the paper, "ensemble"). i.e., in order to compare two models one would have to compare them on the same number of sampled trajectories or the metric would be invalid otherwise.”**
>
> This is true. In practice, these would have to be defined for a particular benchmark, such as the parameters $p$ for the energy score or the variogram score, or $\alpha$ for the MSIS. We agree that deriving a guideline for choosing these parameters will be important for practical applicability. However, such a guideline would by necessity be scenario-dependent, since different applications will have vastly different prediction dimensionalities. We plan to investigate this in future work.
>
> **“Finally, the KNN score's intuition is that the best model is able to place a forecast close in L2 distance to the ground truth. Note that for a univariate forecast this is equivalent to saying that the k-th best forecast in an ensemble, measured in squared error, has low squared error. Setting k=1, this is equivalent to choosing the model with the "best in hindsight" MSE. In other words, KNNS does not appear to measure the quality of distributional forecasts, a desired property set out in the paper, but only point forecast error. ”**
>
> Evaluating based on a "best in hindsight" MSE is a better option for distributional predictions than the standard MSE, because a method that only produces a point prediction would perform worse than a method that produces multiple predictions based on a distribution. An optimal model under a "best in hindsight" MSE would spread its predictions over the area of possible futures, with more predictions in areas where it expects a higher probability.
>
> **“Multiple random projections sampled and with sufficient latent dimension would be justified for only very high dimensionality tasks such as spatiotemporal forecasting in earth sciences, etc. This doesn't appear to be the case with the experiment setup.”**
>
> Many multivariate time series applications (e.g., stock market or energy forecasting) consist of tens or even hundreds of individual time series with a forecast horizon of about 10-50 time steps. Flattening such a forecast results in forecast vectors of several hundreds to thousands of dimensions, which is far too much for a valid density estimate with a manageable number of samples.
>
> **“The introduction of KNNS is somewhat counterintuitive. The paper first reports that out of sample likelihood (i.e., perplexity) is a desirable metric; but due to its practicality immediately moves to KNN distance. I believe this link should be substantiated.”**
>
> We start with the logarithmic score, which computes the log density of the prediction model at the observation. Then we replace the model density, which is not evaluable for many models, by a KNN density estimate. For $\mathbf x, \mathbf y \in \mathbb R^D$ and if $V_D$ is the volume of the unit sphere in $D$ dimensions, this results in $\log \frac{k}{|\mathcal X|V_D\|\text{NN}_{\mathcal X}^k(\mathbf y) - \mathbf y\|^D}$. Given a certain evaluation scenario, $D$, $k$, $|\mathcal X|$ and $V_D$ are constant, so to simplify the computation we omit them for our score. The result is the negative log distance to the kth-nearest neighbor.
>
> **“Why didn't you consider using the average of the first k distances to decrease variance?”**
>
> We have actually thought about averaging over the closest $k$ neighbors, but we were worried that this would deviate even further from the idea of combining a density estimate with the log score. Instead of computing the score with $k=3$, one could compute the mean distance with $k=5$, or as an alternative the median distance with $k=5$. We would choose the latter, as this corresponds to the original score with $k=3$.

---

### Official Review · Reviewer_Hf7C · 2023-10-30

**Soundness:** 1 poor
**Presentation:** 1 poor
**Contribution:** 1 poor
**Rating:** 1
**Confidence:** 5

**Summary:**

The paper explores the challenge of scoring forecasts in the context of multivariate probabilistic forecasting. In response to limitations found in current scoring methods for multivariate distributions, the authors introduce the K nearest neighbor score, which relies on density estimation. Through comparisons with various existing scores on simulated and real-world datasets, the paper demonstrates the advantages of the new score, both qualitatively and quantitatively.

**Strengths:**

- Studying scoring rules for multivariate distributions holds significant importance in numerous applications. The development of improved scores with enhanced properties is an important topic in machine learning.

- The paper aligns with a recent empirical investigation (Caroll, 2022) that evaluates various scoring rules for multivariate distributions.

- The conducted experiments include both synthetic and real-world datasets.

**Weaknesses:**

- The paper's contributions are not clearly articulated and appear to be quite brief.
	- The proposed method is introduced in Section 3, but it's unclear whether it qualifies as a proper scoring rule. If it does, it's essential to provide a formal proof.
	- The statement, "We draw inspiration from the Energy Score," raises questions, as the log score and energy score are very different scores. Could you provide a theoretical justification for using the log score to evaluate multiple projections? Note that the log score may not be interpretable in this context, and your projections might not yield meaningful results.
	- Your score's definition involves both a density model and a score. Please clarify this relationship.
	- Please include a reference for your energy score proof in the Appendix.
	- The paper's contributions concerning scoring rules for multivariate distributions are unclear. Additionally, a recent and important reference, "Regions of Reliability in the Evaluation of Multivariate Probabilistic Forecasts" (ICML 2023), appears to be missing.
	- There seem to be various approximations and challenges in implementing your method, such as not enforcing orthogonality of rows and the issue of sampling enough prediction vectors due to the curse of dimensionality. It's unclear how these challenges and design choices affect your proposed score.

	`
- The paper requires significant revisions for improved clarity, mathematical rigor, and notations.
	- For example, some specific issues include the distinction between p(X) and P in S(P, y) in Section 2, the undefined notation for P and Q in equation (1), and the unclear meaning of \mathbb{X.
	- Section 2.2 mentions "lower case x" without defining it. Please provide a clear definition.
	- The paper mentions "an ensemble of points X" and later "a set of predictions X." Please use consistent terminology to avoid confusion.
	- The notation "$i \in [1, K]$" implies continuity. Please clarify or use appropriate notation.
	- There is an issue in the denominator of expression (8) that needs correction.
	- It's important to distinguish between a score and a metric, as they are distinct concepts. Please provide a clear explanation.

- The statement, "Since all marginals are evaluated independently, certain properties of the distribution are lost," needs further clarification. If the energy score is a proper scoring rule, explain what specific properties are lost and why.

- The statement, "Only the green one mimics dependencies between time steps correctly," is disputable. Having all realizations in the predictive region does not necessarily imply a correct capture of true uncertainty.

- The assertion, "As a result, these metrics should be avoided when assessing probabilistic predictions," needs further elaboration and support. Clarify under what circumstances these metrics should be avoided and why.

**Questions:**

See weaknesses.

---

> ### Author Response · Authors · 2023-11-20
>
> Thank you for your feedback! We see from the reviews that the paper needs significant revision and would like to use this forum as an opportunity for further clarification and feedback.
>
> **“The paper aligns with a recent empirical investigation (Caroll, 2022)”**
>
> Thanks for pointing out the work by Caroll, 2022. We unfortunately could not find the paper based on the name alone. Could you give us the full reference?
>
> **“it's unclear whether it qualifies as a proper scoring rule. If it does, it's essential to provide a formal proof. [...] Could you provide a theoretical justification for using the log score to evaluate multiple projections?”**
>
> We agree, that a proof of propriety would be ideal, we only measure this property with our experiments. It should be sufficient to prove, that the inner part of the KNN score is a valid density estimate, do you agree? Because replacing $p(x)$ of the strictly proper logarithmic score with a valid density estimate should result in a proper scoring rule.
>
> **“Your score's definition involves both a density model and a score. Please clarify this relationship.”**
>
> We start with the logarithmic score, which computes the log density of the prediction model at the observation. Then we replace the model density, which is not evaluable for many models, with a KNN density estimate. For $\mathbf x, \mathbf y \in \mathbb R^D$ and if $V_D$ is the volume of the unit sphere in $D$ dimensions, this results in $\log \frac{k}{|\mathcal X|V_D\|\text{NN}_{\mathcal X}^k(\mathbf y) - \mathbf y\|^D}$. For a given evaluation scenario, $D$, $k$, $|\mathcal X|$ and $V_D$ are constant, so to simplify the computation we omit them for our score.
>
> **“Please include a reference for your energy score proof in the Appendix.”**
>
> What reference are you missing here? We refer to the proof in Appendix A in the second sentence of Section 2.5.1. The proof is our own work, so there is no reference to another paper.
>
> **“a recent and important reference, "Regions of Reliability in the Evaluation of Multivariate Probabilistic Forecasts" (ICML 2023), appears to be missing”**
>
> Thanks for pointing out this work! It is indeed highly relevant.
>
> **“There seem to be various approximations and challenges in implementing your method, such as not enforcing orthogonality of rows and the issue of sampling enough prediction vectors due to the curse of dimensionality. It's unclear how these challenges and design choices affect your proposed score.”**
>
> We have observed in preliminary experiments that omitting the orthogonality requirement has at most very small effects that are overshadowed by other factors. The number of prediction vectors is one such factor; we generally use 100 vectors, as this is often the default in related work. As these certainly depend on the internal data structure and also on the model prediction: do you think a broad experimental setup with various models and data sets would be sufficient to quantify deviations regarding the impact of omitting the orthogonality of rows and the number of samples?
>
> **“The paper requires significant revisions for improved clarity, mathematical rigor, and notations. ”**
>
> Thanks also for pointing out the problems with the mathematical notation, we really appreciate it! However, $\mathbb{X}$ is defined in the introduction to Section 2.
>
> **“The statement, "Since all marginals are evaluated independently, certain properties of the distribution are lost," needs further clarification. If the energy score is a proper scoring rule, explain what specific properties are lost and why.”**
>
> We agree that this statement is phrased too strongly. Perhaps "underrepresented" or something similar would be a better choice here.

---

> ### Author Response · Authors · 2023-11-20
>
> **“The statement, "Only the green one mimics dependencies between time steps correctly," is disputable. Having all realizations in the predictive region does not necessarily imply a correct capture of true uncertainty.”**
>
> Figure 1 should be an intuitive example of why a scoring rule should not score each point independently. The observed bivariate time series repeats the same values twice, and there is a clear correlation between the two individual series.The implicit assumption, of course, is that this is always the behavior of the true data. Now, if you look at a particular time step of a particular series, each model has exactly the same deviations around the observation, so the uncertainty around each point is the same. This is why scoring rules like CRPS score them all the same. However, assuming that the temporal and inter-series correlation structure shown by the observation is representative of the true data, only the green forecast properly reflects these dependencies. It is much more likely to observe the green forecast than the orange or red forecast.
>
> **“The assertion, "As a result, these metrics should be avoided when assessing probabilistic predictions," needs further elaboration and support. Clarify under what circumstances these metrics should be avoided and why.”**
>
> An ideal probabilistic forecast should match the (unknown) distribution of the data. An optimal model for MSE or MAE does not need to predict a distribution at all, predicting the mean or median is sufficient for an optimal score. For example, if the data distribution has two equal-sized modes, the density at the mean could be zero, but would be optimal if evaluated by MSE.

---

> ### Comment · Reviewer_Hf7C · 2023-11-21
>
> We appreciate the authors' response. However, my evaluation remains unchanged, as I still identify several flaws in the paper.
>
> - "Thanks for pointing out the work by Caroll, 2022. We unfortunately could not find the paper based on the name alone. Could you give us the full reference?"
>
>    - **Alexander, Carol, et al. "Evaluating the discrimination ability of proper multi-variate scoring rules." Annals of Operations Research (2022): 1-27.**
>
>  - "It should be sufficient to prove, that the inner part of the KNN score is a valid density estimate, do you agree?"
>
> 	- **I believe you should use the definition of propriety and demonstrate that the expectation of your score with the optimal prediction is always smaller or equal to any other prediction.**
>
>
> - What reference are you missing here? We refer to the proof in Appendix A in the second sentence of Section 2.5.1. The proof is our own work, so there is no reference to another paper.
>
> 	- **You are not the first one to prove this fact. This is a well-known fact in the scoring rule literature. See, for example, "Energy statistics: A class of statistics based on distances.**
>
>
> - “The assertion, "As a result, these metrics should be avoided when assessing probabilistic predictions," needs further elaboration and support. Clarify under what circumstances these metrics should be avoided and why.”
>
> 	- **It is true that you should not use MSE alone to evaluate probabilistic predictions. However, it is also useful to assess multiple functionals, such as the mean or specific quantiles. Hence, I disagree with this statement.**

---

### Official Review · Reviewer_aR1R · 2023-11-04

**Soundness:** 2 fair
**Presentation:** 2 fair
**Contribution:** 2 fair
**Rating:** 5
**Confidence:** 4

**Summary:**

This paper explores an evaluation metric tailored for probabilistic multivariate time series forecasting, a notable stride within a relatively underexplored domain. It underscores the limitations of existing metrics: CRPS and CRPS-Sum cater to univariate forecasting, Energy Score exhibits insensitivity to correlation differences, and Variogram Score lacks rotation invariance. Pioneering the k-nearest Neighbor (KNN) score grounded in density estimation, the paper eloquently delineates both the qualitative and quantitative merits of the proposed metric.

**Strengths:**

The endeavor to refine evaluation metrics for multivariate time series forecasting is commendable, particularly as this sphere warrants further investigation. The KNN score, premised on density estimation, is presented as a remedy to the issues inherent in existing metrics, offering a novel perspective that could potentially catalyze advancements within this field.

**Weaknesses:**

A critical determinant of the proposed metric's efficacy is the selection of the number of neighbors; however, the paper falls short of providing a rigorous justification for this parameter choice. This omission may hinder the metric's practical adoption within the time series community. Additionally, while employing random projection for dimension reduction, the paper lacks a thorough theoretical analysis concerning the impact of this technique, which could potentially undermine the robustness or interpretability of the findings.

**Questions:**

The KNN method, albeit simplistic in its approach towards density estimation, forms the crux of the proposed metric. How does this method fare when juxtaposed against the Kernel Density Estimation method, especially in terms of accuracy and computational efficiency?

---

> ### Author Response · Authors · 2023-11-20
>
> Thank you for your feedback! We see from the reviews that the paper needs significant revision and would like to use this forum as an opportunity for further clarification and feedback.
>
> **“A critical determinant of the proposed metric's efficacy is the selection of the number of neighbors; however, the paper falls short of providing a rigorous justification for this parameter choice.”**
>
> The number of neighbors and the dimensionality depend strongly on the application. We do not give a guideline for the choice of these parameters, since they may depend not only on the dimensionality of the data, but also on its internal structure. We expect that stable parameter settings can be found for different application scenarios with their specific dimensionality ranges. We will investigate this further in future work.
>
> **“Additionally, while employing random projection for dimension reduction, the paper lacks a thorough theoretical analysis concerning the impact of this technique, which could potentially undermine the robustness or interpretability of the findings.”**
>
> We measure the impact of random projections only through experiments, where it becomes clear that the scoring rule behaves improperly when no projections are used. We assume that this is due to the large dimensionality, so random projections into a subspace are a natural choice. Our experiments show that this increases the discriminative power. You mention that you would expect a theoretical analysis regarding robustness and interpretability. Do you mean quantifying the effects of the number of projections and number of samples? Would experiments measuring the noise be sufficient here?
>
> **“How does this method fare when juxtaposed against the Kernel Density Estimation method, especially in terms of accuracy and computational efficiency?”**
>
> We considered kernel density estimation as an alternative, but found it much harder to choose a “good” kernel bandwidth. An ideal bandwidth would depend heavily on the magnitude of the uncertainty in the observed data, a property that is unknown. If the bandwidth is chosen too low, a model only scores if it hits the observation quite exactly, if it is too high, modes get lost in the forecast. Its a parameter that is related to the width of a mode of the data uncertainty. For KNN density estimation, k is independent of scale, but depends only on the relative size of the modes. A small number of k is able to estimate the density for smaller modes of the prediction, but also becomes more noisy. In our case, with many dimensions and relatively few samples, there are only a few plausible options for k.

---

### Meta-Review · Area_Chair_ow8o · 2023-12-06

**Metareview:**

This paper proposes a new KNN scoring rule for multivariate probabilistic forecasting, motivated by the limitations in current scoring rules around capturing correlation structure across time points and time series. The paper compares its KNN scoring rule, based on density estimation, to other common probabilistic forecasting scoring rules on real and synthetic datasets to showcase its discrimination abilities.

While the problem studied is an important one, reviewers expressed concern about the paper not empirically or theoretically exploring the implementation and design choices enough (such as choice of hyperparameters k,d  and better analysis of the random projection step). Reviewers also complained about inadequate mathematical rigor ( formal proof for kNNS being a proper scoring rule, confounding score and metric terminologies) and a missing discussion around a very relevant related work (Marcotte et al, ICML 23).

This work would benefit from a significant revision and a more thorough empirical study around the design choices of the scoring rule.

**Justification For Why Not Higher Score:**

This work doe not clear the bar for what would be needed for a thorough and detailed empirical evaluation of a new scoring rule as proposed in the paper.

**Justification For Why Not Lower Score:**

N/A

---

### Decision · Program_Chairs · 2024-01-16

Reject